# Communicating social responsibilities through CSR reports: Comparative study of top European and Asia-Pacific airlines

Xing Zhang[1,2]*

1 School of Management, Guizhou University of Commerce, Guiyang, China, 2 Faculty of Business, City University of Macau, Macau, China

* B19092100063@cityu.mo

## Abstract

Recently the outbreak of major social incidents has put the airline industry in the forefront of the debate. However, studies related to the social dimension of the Global Reporting Initiative (GRI) Standards in aviation are limited. To fill the gap, this study explored the social themes of corporate social responsibility (CSR) practices between European and Asia-Pacific based airlines. Quantitative content analysis is employed for comparing the social topics in CSR reports of 20 top airlines from Asia-Pacific and European regions over a 3-year period. It concludes that both regions focused more on labor management relations and supplier assessment. The Asia-Pacific airlines have placed special attention to adequacy of social information provided in their CSR reports while the European airlines kept reporting with a comprehensive method. This paper informs both academics and practitioners on the differences of the social dimension of sustainability between the European and Asia-Pacific aviation industry.

## Introduction

The air transport sector has developed significantly during the last two decades. In 2019, the number of globally airlines carried passengers hit a record high of 4.397 billion, which was approximately threefold that in 1998 [1, 2]. Studies have predicted that this number would have continued to rise had COVID-19 not affected it [3, 4]. Air travel is not only a basic transportation facility, but also a channel that drives the development "of global economic, social, and cultural practices" [5] (p. 1). It has created benefits for the society, such as jobs, global market connections, investments in city facilities, and volunteer events [6]. The recent outbreak of major social crises (e.g., the leakage of customer information in Cathay Pacific Airlines, China East Airlines, and Xiamen Air; the crash accidents of Malaysia Airlines; and the mistreatment of passengers in United Airlines) has violated the Global Reporting Initiative (GRI) social standard and has resulted in negative effects on the global aviation industry. Moreover, these issues have led to an increasing awareness of airlines' corporate social responsibility (CSR) among citizens. Given that this industry has millions of audiences, CSR plays an important role in stakeholders' communication and corporate branding [7]. Thus, airlines must exert

**Competing interests:** The authors have declared that no competing interests exist.

considerable effort on CSR performance under market pressures. As corroborated in a previous study, communicating CSR practices through CSR reports is an efficient approach for engaging with stakeholders [8]. Another study asserted that long-term financial performance can be optimistically affected by communicating the social side of sustainability in the airline industry [9].

Nonetheless, studies related to reporting the social dimension of GRI in aviation remain limited [10], particularly among top airlines in terms of international passenger traffic [11]. To fill this gap, the current work aims to explore the adoption of CSR practices via CSR reports in the social dimension between airlines based in the Asia-Pacific and in Europe.

## Literature review

### Definition and importance of CSR

A study [12] defined CSR as an obligation that companies must fulfill when they make business decisions and take actions to consider benefits to society. Another study [13] proposed that the "social responsibility of business encompasses the economic, legal, ethical, and discretionary expectations that society has of organizations at a given point of time" (p. 500). A previous research [14] claimed that CSR is a morally required corporate behavior that is "alleged by a stakeholder to be expected by society. . . and is therefore justifiably demanded of business" (p. 374). Researchers generally agree that CSR is an obligation that companies must fulfill by considering social values while engaging with stakeholders [10]. In accordance with guidelines of the International Civil Aviation Organization (ICAO) [5], sustainable development has three dimensions: economic, social, and environmental objectives. Effective CSR practices will provide advantages to firms and long-term value to society. In a study [15], CSR indicates the values and mission statements of corporations. Through CSR practices, stakeholders can determine what corporations represent. Previous researches [16, 17] have suggested that improved sustainability contributes to increased profitability and enhances the value of airline companies. Furthermore, airlines regard CSR as a tool for distinguishing themselves from their competitors [18]. CSR helps the formation of airlines' strategies [19]. Undertaking CSR-related activities is a means for service providers to gain customer loyalty and increase the degree of customer satisfaction [20]. Service failures are common in this industry [21], and thus, a favorable CSR perception by passengers may weaken negative effects [22]. New study shows that customers have increasing awareness of their need in CSR and thus advises airlines that make a lot of efforts in CSR to utilize such need. Raising this awareness helps to maximize the impact of brand attitude and brand trust on customers and encourages them to choose more socially responsible airlines [23, 24].

Reporting social issues plays an indispensable role in CSR communication given the recent outbreak of major social incidents that has placed the airline industry at the forefront of debates. On the basis of previous studies, the resilience of the financial performance of airlines can be enhanced by communicating social responsibilities [9]. Meanwhile, an improved performance of social sustainability contributes to the increase in productivity by increasing the job satisfaction and sense of belonging of employees [25].

Consequently, airlines actively engage in building CSR reputation and providing support for social responsibilities. Simultaneously, the public is expecting airlines to undertake socially responsible practices [19].

### CSR reporting as a strategic tool for CSR communication

CSR reporting is an approach adopted by corporations to provide information and make commitments related to sustainable issues to their stakeholders [26]. Studies [27–30] have claimed

that many business organizations consider CSR or sustainability reports as efficient means of communication to improve stakeholders' understanding of a company's actions by portraying their images in a favorable manner [31]. The importance of CSR communication has been recognized in the world of business; thus, studying CSR reports from different regions, such as comparative studies between the UK and Germany [32], the UK and the USA [33], China and India [34], and Australia and Slovenia [26], has become an increasing trend. One study [35] conducted a comparative analysis of the quality of CSR reporting among companies in China, India, Malaysia, and the UK from the perspectives of culture and governance structure. This study found that the disclosure of CSR is partly affected by national culture. Moreover, the quality of CSR reports of UK companies is better than those of the three other countries. A similar pattern was observed in another research [36], which witnessed the CSR disclosure of UK and Hong Kong companies reported a growing trend of CSR reporting in both regions. In particular, UK companies have exhibited a prominent positive trend in CSR communication.

According to a previous study [37], the number of CSR reports produced by airlines was insufficient compared with the size of the industry 10 years ago. In addition, some drawbacks of CSR reporting in the aviation sector, including low credibility [38] and the use of unstable findings [39], have been observed. An increasing trend of the significance of CSR reporting, particularly in the tourism industry [40], has been noted, and air transport plays a crucial role in this sector. Airlines have attempted to implement changes in their CSR communication; one noticeable change shows that environmental reporting is being replaced by comprehensive sustainability reporting [41].

## CSR reporting in Asia-Pacific and European airlines

As reported in the World Air Transport Statistics [11], airlines from the Asia-Pacific and European regions outperformed those from North America, Latin America, the Middle East, and Africa in International Passenger Traffic Ranking. Asia-Pacific airlines ranked first in providing service, and European airlines performed well in achieving the environmental goals of the European Commission Directorate-General for Climate Action [42]. However, the CSR performance of Asia-Pacific airlines was heavily criticized in previous studies. For example, the authors of [43] found that Asian airlines reported less on employee involvement, sustainable development, and local community issues compared with their European counterparts. Research has also suggested that minimal attention was given to economic and social issues in the CSR reporting of Asia-Pacific airlines, while European airlines placed considerable attention [44] on just environmental issues in their CSR reports [37, 41–45]. A recent study found similar practices adopted in the reporting of the economic, environmental, and social dimensions of GRI in airlines based in Europe and the Asia-Pacific [10]. Moreover, detailed investigations of the CSR reporting of social dimensions between top Asia-Pacific and European airlines remain limited. Accordingly, the current study aims to analyze the social standard content reported by 10 Asia-Pacific and 10 European airlines over a period of 3 years (2017 to 2019) to identify trends and differences in the CSR reporting of social topics of top European and Asia-Pacific airlines. This study will provide valuable insights for academics and practitioners into the reporting of social issues of top international passenger traffic providers in the aviation industry. In line with the aforementioned objective, the following research questions are presented:

RQ1a. What was the most reported and least addressed social topics by European airlines during the 3-year period?

RQ2. Were there changes in the reporting of social issues among European and Asia-Pacific airlines over time?

RQ3. Were there differences between European and Asia-Pacific airlines in terms of social issue communication in 2017, 2018, and 2019?

## Methodology

### Sampling

Concerning the research objectives and the specialty of airlines industry, this paper adopts a purposive sampling method. When a researcher is very familiar with his research filed and has a general understanding of the survey to be conducted, purposive sampling can be used to obtain more representative samples. Purposive sampling is commonly applied in situations when there is a small population with large variance, and when the boundary of population could not be defined or researchers have limited time, energy and resources [46]. The method is simple and straightforward to use, conforms to the objectives of research and special requirements, and can sufficiently utilize all known data of the samples. However, the results of purposive sampling can be largely affected by tendency of researchers; a deviated judgmental could easily result in sampling deviation and affect the determination of the population. Based on this situation, in order to give a full play to the active role of purposive sampling, researchers must know clearly about the fundamental features of the population, so as to choose representative and typical samples [47]. In this regard, the paper makes a reference to the Skytrax awards, which is the most representative of its kind in airlines industry.

Skytrax awards are widely recognized benchmarks of airline excellence [48] and highly reputable awards in the aviation industry [49]. We selected 10 Asia-Pacific airlines and 10 European airlines from Skytrax awards' "World's Top 100 Airlines in 2019" (Table 1). These leading airlines are closely monitored by multiple stakeholder groups, and therefore, are highly required to communicate their CSR practices openly for reputation management and sustainable development. Their sustainability/CSR reports, which are important tools for monitoring [45] corporate sustainability, are collected to determine differences and developments in their CSR practices. Special attention will be given to the social dimension in these reports because the recent outbreak of several social incidents has placed the airline industry in spotlight. Depending on the availability and accessibility of these reports, we collected 60 reports of the selected airlines for longitudinal study. As a study [50] claimed that measurements should be repeated three times to achieve a decent longitudinal research, we collected the latest CSR reports from 2017 to 2019 of the selected airlines. A total of 60 reports with 3,952,333 words were obtained in our study.

### Content analysis

Content analysis is widely used in content communication research [8], and thus, quantitative content analysis is applied in this study. Given the recognition of GRI Standards in CSR reporting in the aviation industry and its effective establishment [26], we adopted the 19 social topics from the GRI 400 series [51] as our coding scheme. Then, we utilized NVivo 10, a commonly used data analysis software, to determine the function word frequency of the 19 GRI social topics in the selected airlines' CSR reports. By filtering common and overlapping function words (e.g., guidelines, organizations, and standards) to avoid the inflation of intensity [10], we identified the keywords in each subtopic with a weighting of 0.34 or above (Table 2). Keywords are considered the smallest units that provide the most essential information of the

**Table 1. Selected airlines, rankings, reporting year, and word count of CSR reports.**

| Airline | Country/Region | Rank in Skytrax's "World's Top 100 Airlines in 2019" awards | Year | Word count of report |
|---|---|---|---|---|
| *Asia-Pacific Airlines* | | | | |
| Singapore Airlines | Singapore | 2 | 2017 | 27,202 |
| | | | 2018 | 35,334 |
| | | | 2019 | 55,376 |
| All Nippon Airways | Japan | 3 | 2017 | 67,662 |
| | | | 2018 | 63,749 |
| | | | 2019 | 68,647 |
| Cathay Pacific | Hong Kong | 4 | 2017 | 22,688 |
| | | | 2018 | 28,322 |
| | | | 2019 | 35,186 |
| Qantas | Australia | 8 | 2017 | 52,516 |
| | | | 2018 | 51,594 |
| | | | 2019 | 57,641 |
| Thai Airways | Thailand | 10 | 2017 | 88,599 |
| | | | 2018 | 80,103 |
| | | | 2019 | 94,251 |
| Japan Airlines | Japan | 11 | 2017 | 58,442 |
| | | | 2018 | 59,068 |
| | | | 2019 | 59,152 |
| Garuda Indonesia | Indonesia | 12 | 2017 | 49,973 |
| | | | 2018 | 348,715 |
| | | | 2019 | 371,149 |
| China Southern Airlines | China | 14 | 2017 | 93,148 |
| | | | 2018 | 100,162 |
| | | | 2019 | 95,252 |
| Air New Zealand | New Zealand | 16 | 2017 | 30,185 |
| | | | 2018 | 17,495 |
| | | | 2019 | 16,703 |
| Virgin Australia | Australia | 25 | 2017 | 2,987 |
| | | | 2018 | 2,387 |
| | | | 2019 | 10,777 |
| *European Airlines* | | | | |
| Lufthansa | Germany | 9 | 2017 | 47,916 |
| | | | 2018 | 128,425 |
| | | | 2019 | 136,836 |
| Air France-KLM | France | 23+18 | 2017 | 27,434 |
| | | | 2018 | 1,091 |
| | | | 2019 | 210 |
| Aeroflot | Russia | 22 | 2017 | 110,135 |
| | | | 2018 | 131,598 |
| | | | 2019 | 106,663 |
| Iberia | Spain | 26 | 2017 | 19,525 |
| | | | 2018 | 17,365 |
| | | | 2019 | 15,131 |
| Turkish Airlines | Turkey | 27 | 2017 | 25,880 |
| | | | 2018 | 29,817 |
| | | | 2019 | 82,513 |

*(Continued)*

**Table 1.** (Continued)

| Airline | Country/Region | Rank in Skytrax's "World's Top 100 Airlines in 2019" awards | Year | Word count of report |
|---|---|---|---|---|
| Finnair | Finland | 32 | 2017 | 94,580 |
| | | | 2018 | 19,645 |
| | | | 2019 | 17,454 |
| Norwegian | Norway | 39 | 2017 | 77,178 |
| | | | 2018 | 64,823 |
| | | | 2019 | 66,665 |
| Ryanair | Ireland | 59 | 2017 | 105,620 |
| | | | 2018 | 100,768 |
| | | | 2019 | 103,018 |
| Scandinavian Airlines (SAS) | Denmark | 65 | 2017 | 17,447 |
| | | | 2018 | 17,763 |
| | | | 2019 | 84,481 |
| TAP Air Portugal | Portugal | 76 | 2017 | 50,021 |
| | | | 2018 | 49,654 |
| | | | 2019 | 58,212 |
| | | | **Total** | 3,952,333 |

entire text [52]; hence, we examined the keywords with regard to their intensity of use (i.e., frequency) in all the collected reports by using NVivo 10. Considering the various lengths of the examined reports, all keywords counts were standardized for further statistical analysis by dividing the total keyword count of each subtopic by the total word count of each report.

## Statistical analysis

To answer RQ1a and RQ1b, we presented the percentage of keywords intensity and the mean and standard deviation of the keywords' frequency in each subtopic. To answer RQ2, we calculated the average annual growth rate to identify changes and trends in the reports on social practices over the 3-year period. For RQ3, we performed an independent sampled two-tailed t-test to determine the mean differences of use in the 19 social topics of the CSR reports to compare between European and Asia-Pacific airlines.

## Findings

The first research question inquired about the most popular and unpopular social topics reported by European airlines and Asia-Pacific airlines, respectively. Table 3 shows that "GRI 402-Labor/Management relations" ($M_{2017-2019}$ = 9.44%, SD = 8.11%) was the most reported social topic by European airlines, followed by "GRI 416-Customer Health and Safety" ($M_{2017-2019}$ = 5.83%, SD = 5.01%) and "GRI 414-Supplier Social Assessment" ($M_{2017-2019}$ = 5.65%, SD = 4.92%). On the other hand, the least addressed topic was "GRI 408-Child Labor" ($M_{2017-2019}$ = 1.11%, SD = 0.98%), followed by "GRI 409-Forced or Compulsory Labor" ($M_{2017-2019}$ = 1.14%, SD = 1.02%) and "GRI 407-Freedom of Association and Collective Bargaining" ($M_{2017-2019}$ = 1.80%, SD = 1.57%).

Speaking of Asia-Pacific airlines, Table 4 shows that "GRI 402-Labor/Management relations" ($M_{2017-2019}$ = 11.95%, SD = 3.06%) topped as the social reporting topic. "GRI 414-Supplier Social Assessment" ($M_{2017-2019}$ = 9.25%, SD = 2.30%) and "GRI 413-Local Communities" ($M_{2017-2019}$ = 8.66%, SD = 2.24%) ranked second and third, respectively. Whereas, "GRI 408-Child Labor" ($M_{2017-2019}$ = 1.49%, SD = 0.82%) was the least reported social topic,

**Table 2. Social topics in the GRI 400 series and related keywords.**

| Social topics in the GRI 400 series | Topic-specific keywords from the social topics |
|---|---|
| GRI 401: Employment | employees, employed, working, leave, legally, turnover |
| GRI 402: Labor/Management Relations | international, operations, relations, employment, changes, consultation, significant, notice |
| GRI 403: Occupational Health and Safety | work, health, workers, occupational, injuries, control, ill, hazards, workplace |
| GRI 404: Training and Education | training, education, skills, programs, review, accordance, performance, career, assistance |
| GRI 405: Diversity and Equal Opportunity | diversity, equal, nations, remuneration, bodies, gender, indicators, opportunity |
| GRI 406: Nondiscrimination | discrimination, forms, person, effective, elimination, expected, women |
| GRI 407: Freedom of Association and Collective Bargaining | collective, bargaining, association, freedom, employers, united |
| GRI 408: Child Labor | child, labor/labour, ilo, age, minimum, countries |
| GRI 409: Forced or Compulsory Labor | forced, persons, contractor, compulsory, convention, products |
| GRI 410: Security Practices | security, personnel, party, society, third, conduct |
| GRI 411: Rights of Indigenous Peoples | peoples, indigenous, informed, sustainability, context, cultural, identified |
| GRI 412: Human Rights Assessment | right, human, declaration, agreements, principles, contracts |
| GRI 413: Local Communities | community, engagement, stakeholder, groups, vulnerable, actual |
| GRI 414: Supplier Social Assessment | social, criteria, supply, provide, services, business, relationship |
| GRI 415: Public Policy | political, public, policy, contributions, positions, oecd, designed |
| GRI 416: Customer Health and Safety | safety, customer, period, development, concerning, categories, cycle |
| GRI 417: Marketing and Labeling | marketing, labeling, communication, background, recommendations, responsible |
| GRI 418: Customer Privacy | privacy, data, information, protection, breaches, complaints, substantiated, losses |
| GRI 419: Socioeconomic Compliance | regulations, socioeconomic, area, national, environment, foundation, ability |

Remarks: International Labor Organization is abbreviated as ILO. Organization for Economic Co-operation and Development is abbreviated as OECD.

followed by "GRI 409-Forced or Compulsory Labor" ($M_{2017-2019}$ = 2.14%, SD = 0.11%) and "GRI 407-Freedom of Association and Collective Bargaining" ($M_{2017-2019}$ = 2.67%, SD = 0.56%).

In terms of RQ2, we examined the changes in reporting the social issues among European and Asia-Pacific airlines over time by calculating the AAGR. Fig 1 illustrates that 9 out of 19 social topics communicated by European airlines showing positive AAGR during the 3-year period (GRI 405, GRI 406, GRI 407, GRI 410, GRI 411, GRI 412, GRI 414, GRI 417, GRI 418). Whereas, there were 4 out of 19 topics reported by Asia-Pacific airlines showing positive AAGR (GRI 402, GRI 413, GRI 414, GRI 416). Moreover, the intensity of reporting on GRI 411 by European airline dramatically increased from 2017 to 2019.

Regarding RQ3, we examined the differences between European (EU) and Asia-Pacific (AP) airlines in terms of social issues reporting in 2017, 2018, 2019 accordingly. We employed a two-tailed t-test to compare the means of the reporting intensity between the two regions. Then, we noticed significant differences in the intensity of reporting GRI 402 ($M_{AP}$ = 0.82 vs. $M_{EU}$ = 1.17, t = -2.45, p = 0.025*) and GRI 411 ($M_{AP}$ = 0.76 vs. $M_{EU}$ = 0.20, t = 12.4, p = 0.009**) among the airlines in both regions in 2017 (Fig 2). In 2018, a significance of the

**Table 3. The percentage of European airlines reporting on the social topics (N = 10).**

| Social topics reported by European airlines | 2017 | 2018 | 2019 | Mean | SD |
|---|---|---|---|---|---|
| GRI 401 | 0.04% | 6.49% | 4.68% | 3.74% | 3.33% |
| GRI 402 | 0.07% | 14.27% | 13.97% | 9.44% | 8.11% |
| GRI 403 | 0.02% | 4.06% | 3.66% | 2.58% | 2.23% |
| GRI 404 | 0.03% | 6.09% | 5.22% | 3.78% | 3.28% |
| GRI 405 | 0.02% | 3.46% | 4.14% | 2.54% | 2.21% |
| GRI 406 | 0.01% | 2.88% | 4.01% | 2.30% | 2.06% |
| GRI 407 | 0.01% | 2.43% | 2.95% | 1.80% | 1.57% |
| GRI 408 | 0.01% | 1.90% | 1.41% | 1.11% | 0.98% |
| GRI 409 | 0.01% | 1.99% | 1.41% | 1.14% | 1.02% |
| GRI 410 | 0.02% | 3.41% | 3.48% | 2.30% | 1.98% |
| GRI 411 | 0.01% | 7.04% | 9.70% | 5.58% | 5.01% |
| GRI 412 | 0.02% | 4.26% | 5.24% | 3.17% | 2.77% |
| GRI 413 | 0.04% | 6.30% | 7.34% | 4.56% | 3.95% |
| GRI 414 | 0.04% | 9.26% | 7.64% | 5.65% | 4.92% |
| GRI 415 | 0.02% | 3.33% | 3.12% | 2.16% | 1.85% |
| GRI 416 | 0.04% | 8.67% | 8.77% | 5.83% | 5.01% |
| GRI 417 | 0.02% | 4.71% | 4.74% | 3.16% | 2.72% |
| GRI 418 | 0.02% | 4.99% | 4.69% | 3.23% | 2.79% |
| GRI 419 | 0.02% | 4.47% | 3.84% | 2.78% | 2.41% |

intensity of reporting GRI 412 ($M_{AP}$ = 0.20 vs. $M_{EU}$ = 0.38, t = -2.41, p = 0.027*) was observed between the two regions (Fig 3). As for the intensity of reporting in 2019 (Fig 4), we found out the differences of reporting on GRI 412 ($M_{AP}$ = 0.30 vs. $M_{EU}$ = 0.42, t = -2.36, p = 0.034*) and GRI 414 ($M_{AP}$ = 0.90 vs. $M_{EU}$ = 0.61, t = 2.118, p = 0.048*).

**Table 4. The percentage of Asia-Pacific airlines reporting on the social topics (N = 10).**

| Social topics reported by Asia-Pacific airlines | 2017 | 2018 | 2019 | Mean | SD |
|---|---|---|---|---|---|
| GRI 401 | 5.71% | 6.04% | 5.80% | 5.85% | 0.17% |
| GRI 402 | 8.42% | 13.61% | 13.81% | 11.95% | 3.06% |
| GRI 403 | 5.86% | 5.81% | 5.50% | 5.72% | 0.20% |
| GRI 404 | 5.78% | 6.05% | 6.54% | 6.12% | 0.39% |
| GRI 405 | 10.58% | 3.77% | 3.36% | 5.90% | 4.06% |
| GRI 406 | 4.15% | 2.35% | 2.98% | 3.16% | 0.91% |
| GRI 407 | 3.31% | 2.30% | 2.39% | 2.67% | 0.56% |
| GRI 408 | 2.43% | 1.10% | 0.94% | 1.49% | 0.82% |
| GRI 409 | 2.26% | 2.05% | 2.10% | 2.14% | 0.11% |
| GRI 410 | 3.27% | 3.22% | 3.40% | 3.30% | 0.09% |
| GRI 411 | 7.81% | 6.72% | 6.47% | 7.00% | 0.71% |
| GRI 412 | 4.12% | 2.45% | 3.51% | 3.36% | 0.85% |
| GRI 413 | 6.30% | 10.76% | 8.91% | 8.66% | 2.24% |
| GRI 414 | 6.60% | 10.41% | 10.75% | 9.25% | 2.30% |
| GRI 415 | 5.39% | 2.82% | 3.47% | 3.89% | 1.34% |
| GRI 416 | 5.30% | 9.20% | 9.30% | 7.93% | 2.28% |
| GRI 417 | 4.08% | 3.77% | 3.28% | 3.71% | 0.40% |
| GRI 418 | 3.88% | 3.85% | 4.36% | 4.03% | 0.29% |
| GRI 419 | 4.72% | 3.72% | 3.14% | 3.86% | 0.80% |

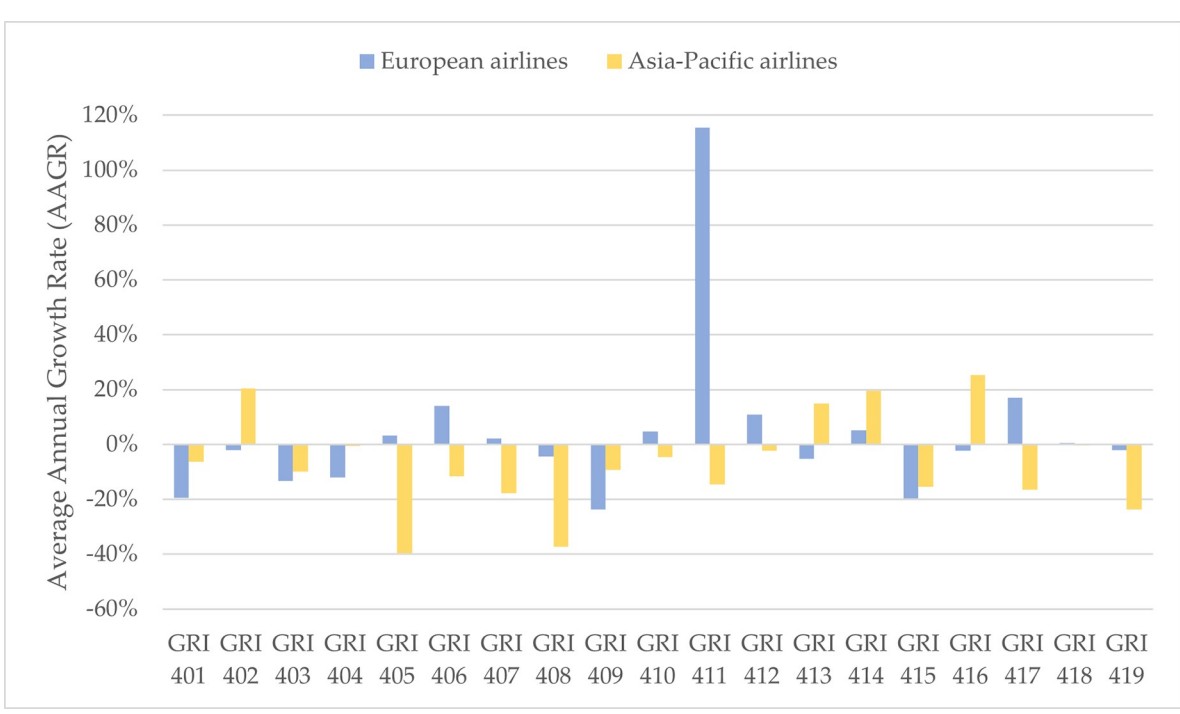

**Fig 1. The average annual growth rate of reporting on social topics among the European and Asia-Pacific airlines from 2017 to 2019.**

## Discussion and conclusions

The current study conducted a comparative and longitudinal analysis of CSR reporting in the social dimension between European airlines and Asia-Pacific airlines. New trends and different practices were disclosed from the analysis of CSR reports.

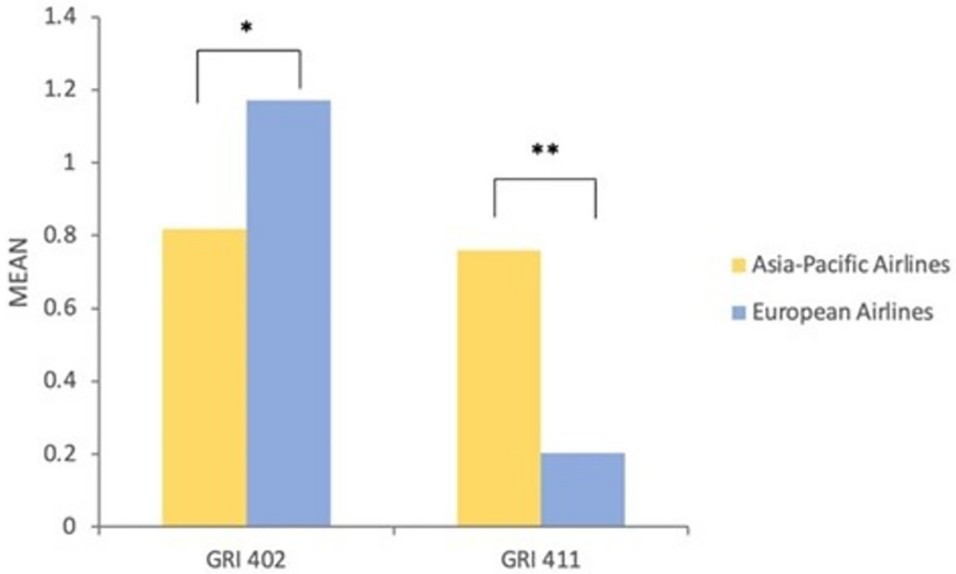

**Fig 2. The differences of communicating in social topics between European airlines and Asia-Pacific airlines in 2017.**

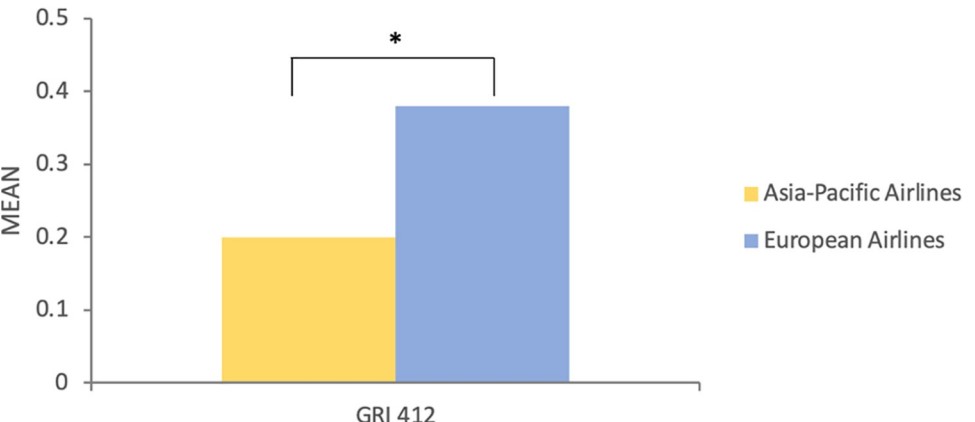

**Fig 3. The differences of communicating in social topics between European airlines and Asia-Pacific airlines in 2018.**

## European airlines favored topics of employment rights though placed focuses on comprehensiveness of social information in their CSR reports

As we discovered that the reporting of 9 social topics, covering diversity, equality, human rights, supplier assessment, and marketing, was increasing with a positive AAGR during the 3-year period. However, European airlines emphasized more on the topics related to employment rights. Congruent with a recent study [8] claimed that the airlines might favor in reporting the topics of employment rights in their CSR reports. The main reason could be a "recent call for balanced sustainable development from the United Nations" [10, 53]. Regarding the topic of rights in employment, a recent research made a deeper study on gender equality in employment. The subject airlines acknowledged that gender inequality could exist when they recruit employees, but they also expressed that they had been seeking for ways to solve such

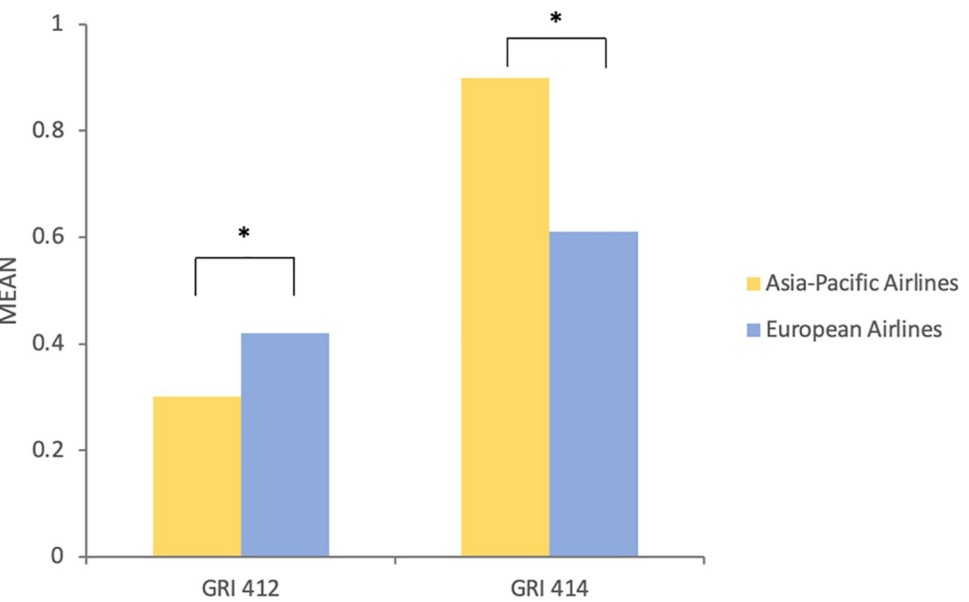

**Fig 4. The differences of communicating in social topics between European airlines and Asia-Pacific airlines in 2019.**

issues. The research gives advice to airlines that the attention of gender equality could be expanded to all functions and posts; it also lays foundation for airlines to rebuild internal rules, regulations and behaviors [54]. Moreover, due to the effective promotion of cross-sectoral CSR practices of governments in the Europe, by setting guidelines, regulations and organizing campaigns for local corporations to follow and participate. The authorities play an important role in providing standards and frameworks in areas like environmental responsibility, health and safety, labor rights [55], etc. For example, on the European Multi-stakeholder Forum, government of Sweden advocated local companies to become the ambassadors of human rights in the world of business [55].

## Salient growth in reporting on social issues by Asia-Pacific airlines

Inconsistent with previous studies [37, 43–45], our findings revealed that Asia-Pacific airlines has made progress in reporting social issues by incorporating adequate social information in their CSR reports from 2017 to 2019. This might be associated with the recent outbreak of social incidents in the airline industry, the increasing demands from stakeholders has put pressure on this industry. In response to the demands, airlines have been improving strategies on CSR communication, particularly in the social dimension [56] to prevent financial losses and enhance branding [55]. Meanwhile, studies show that consumers who have a higher education or income would pay more attention to CSR of airlines. Therefore, it is vital to do enterprise's CSR outreach mainly through such consumer group, which may be an effective strategy to enhance customer loyalty [57]. Another indispensable factor is the active participation of local authorities. Although the recognition of CSR concept in Asia-Pacific countries was later than that in Europe, the Asia-Pacific governments, such as China [58], Japan [45, 59], Australia [26], and South Korea [60] made efforts to advocate CSR practices and communication decade ago. Thus, the growth of CSR communication including the social issues in Asia-Pacific airlines was observed.

## Implications and limitations

As the current research is a unique study on social dimension of CSR communication, it contributes to the body of literature on sustainability communication in the airline sector, by conducting a comparative and longitudinal analysis between two major regions. Practically speaking, this study uncovered the differences and trends between Europe and Asia-Pacific airlines. Therefore, it provides valuable insight to the industry for future decision-making and construction of CSR strategies across regions.

Apart from the contributions, the study has several limitations. The first limitation of this study is the small size of samples. As we selected 20 airlines based in Europe and Asia-Pacific, the performance of reporting practices cannot represent for the whole industry. Further study could present a global view on CSR communication. Furthermore, we investigated CSR reporting from companies' perspective through CSR reports. It is also important to examine the reporting effectiveness from passengers' perspective via adopting surveys.

## Supporting information

**S1 File. The intensity of keywords in GRI 400 series.**
(XLSX)

**S2 File. The word count of the CSR reports Year 2017–2019.**
(XLSX)

## Author Contributions

**Conceptualization:** Xing Zhang.

**Data curation:** Xing Zhang.

**Formal analysis:** Xing Zhang.

**Investigation:** Xing Zhang.

**Methodology:** Xing Zhang.

**Project administration:** Xing Zhang.

**Resources:** Xing Zhang.

**Software:** Xing Zhang.

**Validation:** Xing Zhang.

**Visualization:** Xing Zhang.

**Writing – original draft:** Xing Zhang.

**Writing – review & editing:** Xing Zhang.

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
