## [Decision Letter · Decision Letter 0]

23 Aug 2021

PONE-D-21-20662

Communicating social responsibilities through CSR reports: Comparative study of top European and Asia-Pacific Airlines

PLOS ONE

Dear Dr.Xing Zhang,

Thank you for submitting your manuscript to PLOS ONE. After careful consideration, we feel that it has merit but does not fully meet PLOS ONE’s publication criteria as it currently stands. Therefore, we invite you to submit a revised version of the manuscript that addresses the points raised during the review process.

Please submit your revised manuscript by Oct 07 2021 11:59PM If you will need more time than this to complete your revisions, please reply to this message or contact the journal office at plosone@plos.org. Please include the following items when submitting your revised manuscript:

We look forward to receiving your revised manuscript.

Kind regards,

László Vasa, PhD

Academic Editor

PLOS ONE

Reviewers' comments:

Reviewer's Responses to Questions

**Comments to the Author**

1. Is the manuscript technically sound, and do the data support the conclusions?

Reviewer #1: Yes

Reviewer #2: Yes

2. Has the statistical analysis been performed appropriately and rigorously? 

Reviewer #1: Yes

Reviewer #2: Yes

3. Have the authors made all data underlying the findings in their manuscript fully available?

Reviewer #1: Yes

Reviewer #2: Yes

4. Is the manuscript presented in an intelligible fashion and written in standard English?

Reviewer #1: Yes

Reviewer #2: Yes

5. Review Comments to the Author

Reviewer #1: The article addresses an interesting topic. The literature review contains sufficient amount of items, however, only 10 are from the last 5 years. Since the topic of CSR is relevant and actual, the literature should focus on more resent resources as well. The chosen analysis method is adequate for the presented research questions. The sampling is not described in detail. How were the 10-10 companyies selected from the list? Random sampling?

The conclusions are sound and the authors are aware of the limitations of the research.

Reviewer #2: The paper investigates a rather interesting topic: the CSR activities of the air transportation industry. The approach is comparative (Europe-Asia flagship airlines). The abstract is comprehensive and compact, contains all information needed. Key words are appropriate. The author made a good literature review at a high level, including all theoretical and published recources parts expected. The selected methodology supports the research goal; the authors used the available toolset and datasets well. The outcomes are clear, the results are supported be the methodology. Conclusions are clear and logical, giving useful recommendations based on the findings.

This review was made by the academic editor assigned to this paper.

6. PLOS authors have the option to publish the peer review history of their article (what does this mean?). If published, this will include your full peer review and any attached files.

Reviewer #1: **Yes: **Prof. Kornélia Lazányi

Reviewer #2: No

---

## [Author Response · Author response to Decision Letter 0]

2 Sep 2021

Dear László Vas and reviewers:

Thank you for your letter and the reviewers’ comments on my manuscript entitled " Communicating social responsibilities through CSR reports: Comparative study of top European and Asia-Pacific Airlines " (ID: PONE-D-21-20662). Those comments are very helpful for revising and improving my paper, as well as the important guiding significance to my future research. I have studied the comments carefully and made corrections which I hope meet with approval. In the revised manuscript, the supplementary parts were all highlighted within the document by using red-colored text, and the responds to the reviewers’ comments are as follows.

Replies to the reviewers’ comments:

Reviewer #1:

1. “Since the topic of CSR is relevant and actual, the literature should focus on more resent resources as well.”

Response: Thank you for your advice. I took a thorough check to my paper and add citation of recent studies in the Introduction [3,4], the Literature Review [23,24] and the Discussion and Conclusions [54,57].

2. “The sampling is not described in detail. How were the 10-10 companies selected from the list? Random sampling?”

Response: Thank you for your advice on the sampling part. I have added explanation of sampling method in the revised version (Page 4, Line 126-138).

Reviewer #2:

I really appreciate your recognition and encouragement to my study.

Once again, thank you very much for your constructive comments and suggestions which would help me in depth to improve the quality of the paper.

Your sincerely,

Xing Zhang

Faculty of Business, City University of Macau

School of Management, Guizhou University of Commerce, China

Email: B19092100063@cityu.mo

---

## [Decision Letter · Decision Letter 1]

4 Oct 2021

Communicating social responsibilities through CSR reports: Comparative study of top European and Asia-Pacific Airlines

PONE-D-21-20662R1

Dear Dr. Zhang,

We’re pleased to inform you that your manuscript has been judged scientifically suitable for publication and will be formally accepted for publication once it meets all outstanding technical requirements.

Kind regards,

László Vasa, PhD

Academic Editor

PLOS ONE

Additional Editor Comments (optional):

Reviewers' comments:

Reviewer's Responses to Questions

**Comments to the Author**

1. If the authors have adequately addressed your comments raised in a previous round of review and you feel that this manuscript is now acceptable for publication, you may indicate that here to bypass the “Comments to the Author” section, enter your conflict of interest statement in the “Confidential to Editor” section, and submit your "Accept" recommendation.

Reviewer #2: All comments have been addressed

2. Is the manuscript technically sound, and do the data support the conclusions?

Reviewer #2: Yes

3. Has the statistical analysis been performed appropriately and rigorously? 

Reviewer #2: Yes

4. Have the authors made all data underlying the findings in their manuscript fully available?

Reviewer #2: Yes

5. Is the manuscript presented in an intelligible fashion and written in standard English?

Reviewer #2: Yes

6. Review Comments to the Author

Reviewer #2: The authors accepted my previous recommendations and improved they paper well. In its current status it reaches the quality expectations of the journal. The findings can contribute to the existing knowledge and would be of interest of the PLOS ONE readers.

7. PLOS authors have the option to publish the peer review history of their article (what does this mean?). If published, this will include your full peer review and any attached files.

Reviewer #2: No

---

## [Editor Report · Acceptance letter]

7 Oct 2021

PONE-D-21-20662R1 

Communicating social responsibilities through CSR reports: Comparative study of top European and Asia-Pacific Airlines 

Dear Dr. ZHANG:

I'm pleased to inform you that your manuscript has been deemed suitable for publication in PLOS ONE. Congratulations! Your manuscript is now with our production department. 

Kind regards, 

on behalf of

Prof. Dr. László Vasa 

Academic Editor

PLOS ONE